# Biostimulants Enhance the Nutritional Quality of Soilless Greenhouse Tomatoes

**DOI:** 10.3390/plants13182587

**Published:** 2024-09-15

**Authors:** Hayriye Yildiz Dasgan, Kahraman S. Aksu, Kamran Zikaria, Nazim S. Gruda

**Affiliations:** 1Department of Horticulture, Faculty of Agriculture, University of Cukurova, Adana 01330, Turkey; sm.aksu87@icloud.com (K.S.A.); muhammadkamran7253@gmail.com (K.Z.); 2Institute of Plant Sciences and Resource Conservation, Division of Horticultural Sciences, University of Bonn, 53113 Bonn, Germany

**Keywords:** antioxidants, fruit quality, hydroponics plant growth, produce quality, *Solanum lycopersicum* L., yield

## Abstract

The application of biostimulants in vegetable cultivation has emerged as a promising approach to enhance the nutritional quality of crops, particularly in controlled environment agriculture and soilless culture systems. In this study, we employed a rigorous methodology, applying various biostimulants amino acids, Plant Growth-Promoting Rhizobacteria (PGPR), fulvic acid, chitosan, and vermicompost along with mineral fertilizers, both foliar and via the roots, to soilless greenhouse tomatoes during spring cultivation. The experiment, conducted in a coir pith medium using the ‘Samyeli F1’ tomato cultivar, demonstrated that plants treated with biostimulants performed better than control plants. Notable variations in nutritional components were observed across treatments. PGPR had the best effects on the physical properties of the tomato fruit, showing the highest fruit weight, fruit length, equatorial diameter, fruit volume, fruit skin elasticity, and fruit flesh hardness while maintaining high color parameters L, a, and b. PGPR and fulvic acid demonstrated significant enhancements in total phenolics and flavonoids, suggesting potential boosts in antioxidant properties. Amioacid and vermicompost notably elevated total soluble solids, indicating potential fruit sweetness and overall taste improvements. On the other hand, vermicompost stood out for its ability to elevate total phenolics and flavonoids while enhancing vitamin C content, indicating a comprehensive enhancement of nutritional quality. In addition, vermicompost had the most significant impact on plant growth parameters and total yield, achieving a 43% increase over the control with a total yield of 10.39 kg/m^2^. These findings underline the specific nutritional benefits of different biostimulants, offering valuable insights for optimizing tomato cultivation practices to yield produce with enhanced health-promoting properties.

## 1. Introduction

Tomato (*Solanum lycopersicum*) is a herbaceous species in the *Solanaceae* family. Global tomato production reached approximately 187 million metric tons in 2020, making it one of the most widely cultivated crops worldwide. China is the largest producer, with around 65 million metric tons, India and Türkiye are the second and third largest producers, with 20.5 million and 13.2 million metric tons, respectively [1]. Tomato fruits are nutrient-rich, offering essential vitamins such as A, C, and K, folate, and fibers. Their high lycopene content is well recognized for its antioxidant properties, which protect the human body from damage caused by free radicals. Low in calories and high in water, tomatoes support hydration and weight management, while their fiber aids digestion. Overall, tomatoes are a valuable addition to a healthy diet, providing various health benefits [2,3,4].

Soilless systems represent a promising agricultural advancement, offering greater efficiency and reliability. This modern technique is gaining global popularity for addressing challenges such as limited arable land, water scarcity, and climate constraints [5,6]. Vegetables are grown in nutrient solutions without soil, allowing precise environmental control, leading to resource efficiency, year-round cultivation, and increased yields. The economic benefits of soilless tomato cultivation are notable due to higher productivity and effective resource use [7,8]. With proper management, both product quality and overall output can be optimized, boosting local economies and farmers’ incomes [9].

Biostimulants encompass a range of substances [10], such as amino acids, fulvic and humic acids, seaweed and plant extracts, inorganic compounds, beneficial bacteria, beneficial fungi, chitosan and chitosan-like polymers [11], and vermicompost [12]. These materials can be applied through seed coating, pelleting, root application, and foliar application [13,14].

Recently, biostimulants have gained significant attention in agriculture for their innovative, environmentally friendly technologies that address critical challenges without adverse environmental impacts [15,16,17]. According to the European Biostimulants Industry Council (EBIC), biostimulants are substances or microorganisms applied to plants or the rhizosphere to stimulate natural processes, enhancing nutrient uptake, efficiency, tolerance to abiotic stresses, and crop quality. Research highlights their role in promoting root development, improving nutrient uptake efficiency, and increasing plant resilience to abiotic stress, e.g., salinity [18,19,20]. As agricultural practices evolve toward sustainability and reduced reliance on synthetic inputs, biostimulants present a promising approach for fostering healthier, more resilient crops [21].

In floating hydroponic culture, it has been reported that biostimulants such as PGPR, mycorrhiza, and microalgae reduce the use of mineral fertilizers in green leafy vegetables such as lettuce, basil, and spinach [22,23,24,25], as well as in soilless-grown capia red peppers with coir [26]. These biostimulants are environmentally friendly practices that enhance product quality, plant growth, and yield.

Amino acids, including structural proteins such as glutamate, histidine, proline, and glycine betaine, are often deficient in plant structures but play a crucial role in protecting plants from abiotic stresses and stimulating physiological processes through signaling [27,28,29]. Known as “protein hydrolysates”, amino acids serve multiple functions in plants: they act as stress-reducing agents, sources of nitrogen, and precursors to hormones. Additionally, amino acids are precursors or activators of phytohormones and growth substances [30,31,32].

Fulvic acids are soluble organic compounds found in nature, distinguished by functional groups such as carbonyl, carboxyl, hydroxyl, phenolic hydroxyl, and quinone, which enable them to chelate and exchange ions [33,34]. Carboxyl groups’ high cation exchange capacity allows better cation absorption than humic acids [35]. The small molecular weight of fulvic acids facilitates their use as trace element synergists or plant growth regulators, often applied through foliar fertilization in vegetable production [36,37]. This small size also enables easy passage through cell membranes, enhancing the transport and availability of iron and other micronutrients [38]. Consequently, fulvic acids increase chlorophyll content, nitrogen use efficiency, and photosynthetic rate.

Plant Growth Promoting Rhizobacteria (PGPR), also known as probiotic rhizobacteria, offers significant benefits to both the growing medium and plant health. PGPR supports plant development through several mechanisms: breaking down heavy metals, producing hormones, fixing nitrogen in the root zone, enhancing mineral and water uptake, promoting root growth, and increasing enzyme activity [39]. Enriching the root zone with nitrogen fixation and the mineralization of potassium and phosphorus, these probiotic bacteria enhance overall plant growth [40,41,42].

Chitosan is a biopolymer derived from chitin, found in the exoskeletons of crustaceans such as shrimp, crabs, and lobsters, as well as in the cell walls of fungi. This versatile compound holds significant potential for enhancing crop output. Chitosan is involved in plant defense mechanisms by stimulating resistance to pathogens and pests. It promotes plant growth and improves seed coating, protecting against abiotic stress damage [20]. Additionally, chitosan exhibits chelating properties, facilitating the availability of essential nutrients, and contains nitrogen, which contributes to soil fertility and plant nutrition [43].

Vermicompost, produced from organic waste processed by worms, can be applied directly to soil or plant leaves. It is rich in essential macro- and micronutrients and millions of beneficial microorganisms. Vermicompost significantly enhances overall plant growth, promotes the development of new shoots and leaves, and improves both produce quality and shelf life. It also increases plant resistance to pests and diseases and abiotic stress [20,44,45]. Additionally, vermicompost enhances soil structure, aeration, and water retention and helps prevent soil erosion. It enriches the soil with beneficial microorganisms, such as nitrogen fixers, phosphorus solubilizers, and cellulose decomposers, while boosting the population and activity of earthworms. Free from pathogens, toxic elements, and weed seeds, vermicompost contains valuable vitamins, enzymes, and plant hormones such as auxins and gibberellins [46].

Adopting innovative biostimulants in soilless-grown tomatoes represents a significant advancement in agriculture, transitioning from traditional soil-based cultivation methods [34,47,48]. Derived from natural and eco-friendly sources, these biostimulants support sustainable practices in soilless systems by enhancing plant growth and yield [49,50,51,52]. Although much research has focused on improving growth and yield through environmental adjustments, there is limited investigation into how biostimulants affect the nutritional quality of tomatoes in soilless systems.

Given the critical importance of tomato quality for human health and overall produce value, this research aims to address this gap. We hypothesize that specific biostimulants can improve the physical fruit properties and enhance the nutritional quality of tomatoes without compromising yield. By identifying these biostimulants, the study seeks to optimize plant performance and nutritional quality, advancing our understanding of their potential to improve produce quality in soilless tomato cultivation.

## 2. Materials and Methods

The trial was conducted in the spring season of 2022, using a 500 m^2^ glasshouse at Cukurova University, Türkiye (36° 59′ N, 35° 18′ E, and 23 m above sea level). The tomato variety “Samyeli F_1_”^®^, known for its favorable physiological and morphological traits during the spring–summer season, was obtained from Anamas Seed Company Ltd. (Antalya, Türkiye) The growing media consisted of polythene-packed coconut coir substrates with dimensions of 100 cm × 20 cm × 4 cm. Four tomato plants were grown on each coco coir pith slab.

### 2.1. Biostimulants Used in This Experiment

The experiment consisted of six treatments, including one control and five biostimulants:T1: ControlT2: Amino acidT3: PGPRT4: Fulvic acidT5: ChitosanT6: Vermicompost

The first biostimulant, “Amino Gold”^®^, is an amino-acid-based product the Teos Tarim company manufactured. “Amino Gold”^®^ amino acid contains 70% total organic matter, 14% organic carbon, 3% organic nitrogen, and 29% free amino acids. The second biostimulant, “Sacaka”^®^, is a commercially available powdered fulvic acid provided by the “Köklü Group” company (Mersin, Türkiye). “Sacaka WS”^®^ fulvic acid comprises 80% total organic matter and 70% fulvic acid. The third biostimulant, “Rhizofil”^®^, is a mixture of beneficial bacteria (PGPR) comprising three species from the NG-Biyoteknoloji company (Istanbul, Türkiye). “Rhizofill”^®^ PGPR biostimulant consisting of a mixture of *Bacillus subtilis* (1 × 10^9^ CFU mL^−1^), *Bacillus megaterium* (1 × 10^9^ CFU mL^−1^), and *Pseudomonas fluorescens* (1 × 10^10^ CFU mL^−1^). The fourth biostimulant, “Nanowet”^®^, is a chitosan-based product containing 2.5% N-Acetyl-d-Glucosamine and 2-acetamide-2-deoxy-β-d-glucose monomers linked by β-1,4 bonds, produced by the Adaga company (Antalya, Türkiye). Finally, the fifth biostimulant used was “Ekosolfarm”^®^ vermicompost, derived from red California worms (*Eisenia foetida*) and produced by the Ekosolfarm company (Manisa, Türkiye). “EkosolFarm”^®^ liquid vermicompost contains 35% total organic matter, 20% humic-fulvic acid, 1.2% nitrogen, 1–2% P_2_O_5_, and 1.5–2.5% K_2_O.

### 2.2. Plant Growing Conditions

The experiment was conducted in a glasshouse with temperatures maintained between 18 and 20 °C at night and between 23 and 28 °C during the day, from March to July. Each treatment consisted of 4 replications comprising 16 plants, resulting in a density of 3.38 plants m^−2^ and a 90 cm × 25 cm spacing between plants (Figure 1). Biostimulant applications commenced 15 days after transplanting and continued for 125 days post-transplant. The tomato plants were supported with ropes, and pollination was facilitated using bumblebees (*Bombus terrestris*). Tomato seedlings were transferred to coco coir slabs on 10 March 2022, with biostimulant applications beginning on 25 March 2022. The first tomato harvest occurred on 2 June 2022, followed by six fruit harvests (Figure 2). The experiment was concluded on 7 July 2022. Biostimulants were applied via both foliar and root methods, with applications occurring 11 times at 10-day intervals. The concentrations of the various biostimulants applied through both root and foliar applications to the plants are shown in Table 1.

### 2.3. Plant Nutrition

Two tanks of nutrient solution, stock A and stock B, were prepared and then diluted together in a single tank with a capacity of 1000 L (Table 2). The nutrient solution was delivered using a drip irrigation system with emitters releasing 1.5 L per hour at the base of each plant. The pH and EC values of the nutrient solution were maintained between 5.5 and 6.0 and between 2.0 and 3.0 dS m⁻^1^, respectively. These pH and EC were adjusted according to the vegetative and reproductive stages of the plants. The tomato plants were grown with the following nutrient solution [47,48] (in mg L^−1^): NO_3_-N (135–225), NH_4_-N (15–25), P (40–50), K (200–400), Ca (150–180), Mg (50–75), Fe (2.8–5.0), Mn (0.8–1.0), Cu (0.3–0.4), Zn (0.3–0.4), B (0.3–0.4), and Mo (0.05–0.1).

### 2.4. Plant Growth Measurements

At the end of the experiment, 120 days after transplanting, plant height, leaf number, stem diameter, and leaf area were measured. The pruned leaves’ weight and area were recorded during the cultivation period. The stem diameter was measured in millimeters using a digital caliper. The number of leaves per plant was recorded, and leaf area was measured with a leaf area meter (Li-3100, LICOR, Lincoln, NE, USA) and expressed in square centimeters per plant. Chlorophyll content was evaluated using a SPAD chlorophyll meter (SPAD-502, Minolta, Osaka, Japan). The leaves’ fresh weight (FW) was recorded before drying them at 65 °C for 24 h. The leaves were then reweighed to determine the dry weight (DW), and the percentage of dry matter content was calculated using the formula DW = 100 × DW FW^−1^ [20].

### 2.5. Fruit Harvest and Measurement of Fruit Properties and Quality Attributes

Tomato fruits were harvested weekly upon reaching the red maturity stage (Figure 3). This study harvested up to 7–8 fruit clusters from the indeterminate greenhouse tomato plants. The cumulative yield was calculated as kg m⁻^2^ for the total harvest. For fruit quality measurements, 15 fruits per replication were sampled during the second harvest. The physical quality properties assessed included fruit weight, equatorial diameter, height, volume, flesh firmness, skin elasticity, and color characteristics (L, a, b) of the fruit skin [48]. Fruit equatorial diameter and height were measured using a digital caliper. Fruit volume was determined by measuring the volume of water displaced by submerging the fruit in a water-filled container. The elasticity of the tomato fruit skin was assessed while the skin was intact, while flesh firmness was measured after peeling the skin using a digital penetrometer (Bareiss HPE-III-Fff, ABQ Industrial, The Woodlands, TX, USA). The fruit skin’s L, a, and b color values were digitally recorded using a portable handheld color spectrophotometer (HunterLab, Reston, VA, USA). Additionally, tomato fruit chemical and antioxidant properties such as pH, electrical conductivity (EC), total soluble solids (TSS), titratable acidity, total phenolics, total flavonoids, and vitamin C content were measured in the tomato fruit.

### 2.6. Determination of Total Soluble Solids, Titratable Acidity, EC, and pH in Tomato Fruits

Total soluble solids (TSS) and titratable acidity were measured from tomato fruit juice using a digital device (Atago PR-101, Tokyo, Japan) [48]. The tomato fruit’s electric conductivity (EC) and pH were measured using pH and EC meters (WTW pH/Cond 3320, Weilheim, Germany) [48].

### 2.7. Determination of Antioxidants in Tomato Fruits

The total phenolic content was determined using a modified approach based on the methodology outlined by Spanos and Wrolstad [53]. The total phenolics extracted were quantified in milligrams of gallic acid (GA) equivalents by measuring absorbance at 765 nm with a UV–visible spectrophotometer (UV-1700 Pharma Spec Shimadzu, Kyoto, Japan). Total flavonoid content in the tomato fruit samples was quantified following the method described by Quettier et al. [54], using the same UV–visible spectrophotometer at 765 nm. Flavonoid concentrations were determined against a calibration curve prepared with standard solutions. Vitamin C levels were measured using the procedure adapted from Elgailani et al. [55]. The tomato fruit was homogenized with a high-speed blender, and 5 mL of the extract was mixed with 45 mL of 0.4% oxalic acid and then filtered. The filtrate was analyzed by combining 1 mL of extract with 9 mL of 2,6-dichlorophenolindophenol sodium salt, and the transmittance was recorded at 520 nm using a UV spectrophotometer.

### 2.8. Statistical Analysis

The data obtained from the experiment were analyzed for variance using the JMP statistical package (version 7.0, SAS Institute, Cary, NC, USA, 2007). Parameters statistically significant at the *p* < 0.05 level were further analyzed. Differences between treatments were assessed using the Least Significant Difference (LSD) multiple comparison test, and evaluations were made accordingly. In addition, all the independent variables were subjected to multiple variable analyses by Pearson correlation matrix ClustVis software (https://biit.cs.ut.ee/clustvis/, accessed on 5 August 2024).

## 3. Results

### 3.1. Effects of the Biostimulants on Plant Growth

Statistically significant differences in plant growth parameters were observed across various treatments (Table 3). Plant height increased by 7.65%, 5.18%, and 4.82% with fulvic acid, vermicompost, and chitosan, respectively, compared to the control. Additionally, the number of leaves increased by 7.69% with vermicompost. Leaf area also showed a substantial increase, with a 74.38% rise in the vermicompost, 73.03% with amino acids, and 60.78% with bacteria compared to the control. Furthermore, stem diameter exhibited notable increases of 10.23% and 8.96% in the chitosan and vermicompost, respectively, compared to the control. Vermicompost significantly increased leaf dry matter to 13.14%, a 21.8% improvement over the control (10.79%). Amino acids (11.55%) and PGPR (11.01%) also showed moderate increases of 7.1% and 2.0%, respectively. Fulvic acid had a minimal effect, raising the dry matter by just 0.8%, while chitosan slightly decreased it to 10.39%. The biostimulants positively influenced chlorophyll content in tomato leaves (Table 3). Vermicompost resulted in the highest chlorophyll content with a SPAD of 53.96, significantly higher than all other treatments. Chitosan followed with a SPAD of 45.70, significantly higher than the control and fulvic acid but lower than vermicompost. PGPR and amino acids improved chlorophyll content with SPAD values of 44.90 and 44.00, respectively. Both treatments show a significant increase compared to the control. Fulvic acid yielded a SPAD of 41.42, higher than the control but lower than the other treatments mentioned. The control had the lowest SPAD at 37.13, indicating the minor chlorophyll content among all treatments.

### 3.2. Effect of Biostimulants on Tomatoes Fruit Color Properties

Table 4 displays the effects of different biostimulants on tomato fruit color parameters, measured as L* (lightness), a*, and b*. L* indicates how light or dark the color is. Control tomatoes had the highest lightness (40.94), suggesting a lighter color than other treatments. Fulvic acid produced the lightest tomatoes (35.41), making them appear darker. a* (red–green axis) measures the red–green spectrum. Vermicompost led to the most intense red color (30.32), significantly higher than all other treatments, indicating a more vibrant red hue. Fulvic acid slightly increased red compared to the control but was less pronounced than vermicompost. b* (yellow–blue axis) reflects the yellow–blue spectrum. Vermicompost also had the highest b* (39.92), indicating a more robust yellow hue than other treatments. This contrasts with the control, which had the lowest b* (32.45), reflecting a less intense yellow. Overall, vermicompost enhanced red and yellow hues in tomatoes, leading to a more vibrant and visually appealing fruit color. Other treatments, like amino acid and PGPR, also improved color parameters but not as significantly as vermicompost.

### 3.3. Effects of Biostimulants on Physical Properties of Tomato Fruits

Table 5 shows the impact of various biostimulants on tomato fruits’ physical and visual properties, including fruit weight, length, diameter, and volume. The statistical analysis confirms that the differences among treatments are significant. PGPR resulted in the heaviest fruits at 257.49 g, significantly outperforming all other treatments. Amino acid, fulvic acid, and chitosan also increased fruit weight compared to the control, with weights ranging from 187.31 g to 189.74 g. The control had, with 164.74 g, the lowest fruit weight. PGPR again stood out, producing the longest fruits at 47.07 mm. Interestingly, vermicompost produced the shortest fruits at 30.36 mm. Like fruit length, PGPR led with 73.23 mm to the most significant diameter. In contrast, vermicompost had the most minor fruit equatorial diameter at 47.45 mm, with a higher number of small fruits.

Amino acids, fulvic acid, and chitosan showed moderate improvements in fruit volume, with values ranging from 168.37 m^3^ to 173.38 m^3^. The control had the lowest fruit volume at 146.62 m^3^. Consistent with the other parameters, PGPR resulted in the most significant fruit volume at 228.45 cm^3^, reflecting its impact on increasing fruit weight and overall fruit size.

The highest skin elasticity was observed in fruits biostimulated with PGPR, fulvic acid, amino acid, and vermicompost. These treatments significantly increased elasticity compared to the control and chitosan. Vermicompost led to the highest flesh firmness, indicating firmer fruit flesh than other treatments. PGPR and amino acid also improved flesh hardness, significantly increasing compared to the control and chitosan.

### 3.4. Impact of Biostimulants on the Nutritional Properties of Tomato Fruits

Figure 4 shows the effects of biostimulants on fruit TSS, titratable acidity, EC, pH, total phenolic, total flavonoid compounds, and vitamin C. The application of various biostimulants significantly influenced the TSS content in tomato fruits (Figure 5). Among the treatments, the highest TSS was in the amino acid, with a mean TSS of 4.96%. This represented a substantial 27.18% increase compared to the control. Vermicompost also improved notably, increasing the TSS to 4.76%, corresponding to a 22.05% increase over the control. The chitosan and fulvic acid exhibited similar effects, with TSS of 4.56% and 4.53%, respectively. These correspond to 16.92% and 16.15% increases compared to the control. The PGPR yielded the lowest TSS increase among the biostimulants, with a TSS of 4.36%, reflecting an 11.79% increase over the control.

Biostimulants generally increased titratable acidity; however, chitosan reduced it. PGPR resulted in the highest TA at 2.97%, representing a 160.5% increase compared to the control (1.14%). Vermicompost also showed a significant increase, with a TA of 2.45%, corresponding to a 114.9% increase. Fulvic acid and amino acid increased TA by 45.6% and 28.9%, respectively, relative to the control.

Vermicompost resulted in the highest EC of 1548 µS cm⁻^1^, representing a **65.7% increase** compared to the control. **PGPR** yielded an EC of 1225 µS cm⁻^1^, a **31.2% increase** relative to the control. **Compared to the control, chitosan, amino acid, and fulvic acid achieved EC increases of 17.1%, 10.1%, and 6.0%, respectively.** The control had the lowest EC of 934 µS cm⁻^1^.

Fulvic acid, amino acid, and chitosan resulted in slightly higher pH values (4.57, 4.55, and 4.51, respectively) than the control (4.48). However, these increases are not statistically significant. Vermicompost and PGPR, on the other hand, led to lower pH values (4.35 and 4.33, respectively) compared to the control, which may indicate increased acidity.

The biostimulants significantly affected the total phenolic content in tomato fruits. PGPR resulted in an 88.74% increase compared to the control. Vermicompost showed a 36.31% rise. Applying fulvic acid, chitosan, and amino acid enhanced the phenolic content by 32.22%, 27.48%, and 18.64%, respectively.

The application of the biostimulants significantly influenced the total flavonoid content in tomato fruits. Fulvic acid resulted in a remarkable increase of 217.84% compared to the control. Amino acid showed a 173.59% increase. Vermicompost reflected a 121.59% rise. PGPR and chitosan contributed to increases of 46.91% and 44.32%, respectively.

The impact of various biostimulants on vitamin C content in tomatoes revealed significant differences among the treatments. The highest vitamin C concentrations were observed in the chitosan (19.96 mg) and vermicompost (19.75 mg). These treatments significantly increased the fruit’s vitamin C content compared to the control and other biostimulants. The amino acid, PGPR, and fulvic acid improved vitamin C content to a lesser extent. The control had the lowest vitamin C content at 18.05 mg.

### 3.5. Heat Map Analysis of Biostimulant Influences on Tomato Quality and Nutritional Properties

The heat map analysis provides a comprehensive overview of the effects of various biostimulants on the physical properties of fruit size and nutritional and antioxidant parameters (Figure 5). PGPR emerges a significant positive impact on fruit size parameters, including fruit weight, diameter, length, and volume, as indicated by the darker red shades. This suggests that PGPR is particularly effective in increasing fruit size, thus enhancing the physical attributes of the fruit. Additionally, PGPR also significantly improves fruit skin elasticity and flesh firmness. Furthermore, the PGPR positively influences the nutritional properties of tomato fruit, such as total phenolics, titratable acidity, and EC. While vermicompost lags behind PGPR in terms of tomato fruit size, it excels in other attributes such as flesh firmness, color values a* and b*, electrical conductivity (EC), vitamin C content, and titratable acidity. Amino acids stand out, particularly in enhancing tomato fruit attributes such as total soluble solids and total flavonoids, while having notable positive effects on fruit flesh firmness, fruit skin elasticity, and color value a*. Fulvic acid and chitosan stand out for their high values in total flavonoid and vitamin C content in tomato fruits.

### 3.6. Effects of the Biostimulants on Total Tomato Yield

Statistical analysis indicates significant differences among the treatments (Figure 6). Vermicompost achieved the highest yield at 10.72 kg m^−2^, representing a substantial 43% increase compared to the control (7.50 kg m^−2^). Amino acids produced a 9.31 kg m^−2^ yield, a 24% increase over the control. Fulvic acid yielded 8.58 kg m^−2^, 14% higher than the control. Chitosan yielded 8.26 kg m^−2^, showing a 10% increase relative to the control. PGPR produced 8.07 kg m^−2^, reflecting a 7.6% increase compared to the control. Vermicompost was the most effective treatment, significantly boosting tomato yield, followed by amino acids. All treatments showed improvements over the control, with varying degrees of effectiveness.

### 3.7. Heat Map Analysis of Biostimulant Influences on Tomato Plant Growth, Yield

The heat map analysis reveals that different biostimulants have varied effects on plant growth parameters and tomato yield (Figure 7). Vermicompost stands out as the most successful, particularly in enhancing the leaf dry matter, chlorophyll, leaf number, leaf area, and total yield, where it shows the highest impact, as indicated by the deep red coloration. Amino acid also shows significant positive effects, particularly on leaf number, leaf area, leaf dry matter, and total yield. However, PGPR, fulvic acid, and chitosan exhibited less favorable effects on the measured growth parameters, as indicated by the more neutral or light blue colors. In contrast, the control, indicated by blue shades in the growth parameters and yield, exhibited lower values than the biostimulants.

## 4. Discussion

Agriculture faces immense pressure due to the growing population’s food demands, the environmental impact of excessive conventional fertilizer use, and the challenges posed by climate change, which expose crops like tomatoes to extreme conditions. These factors have significantly affected crop production and quality [56]. New agronomic strategies are being developed to address these challenges and advance sustainable agriculture, with biostimulants emerging as a promising solution [57,58]. However, the question arises whether these biostimulants can also enhance nutritional quality.

### 4.1. Effects of Biostimulants on Plant Growth and Yield of Tomato Plant

In our study, all biostimulants used resulted in better growth and higher fruit production than control plants. Vermicompost emerged as the most effective biostimulant for enhancing tomato plants’ growth parameters and fruit yield (Table 3, Figure 5). Numerous studies have consistently shown that vermicompost significantly boosts plant growth and productivity [59,60,61]. These benefits are primarily attributed to its rich nutrient content, improved soil structure, and the presence of beneficial microorganisms. Our findings align with Truong et al. [62], Ahmadpour et al. [63], Qasim et al. [64], and Tikoria et al. [65], who reported similar increases in vegetative growth and yield in tomato plants treated with vermicompost. Studies have shown that incorporating vermicompost into the root medium enhances macronutrient levels, nutrient uptake, plant performance, and overall biomass.

Vermicompost improves root zone aeration, water retention, microbial activity, and nutrient availability, creating an optimal environment for root development. The presence of humic acids, growth-promoting hormones, and enzymes such as chitinases, amylases, lipases, and cellulases in vermicompost aids in organic matter degradation and nutrient release, making them readily available to plant roots [66]. This stimulates root elongation and enhances nutrient uptake efficiency, leading to robust plant growth and higher productivity. Additionally, vermicompost is rich in beneficial bacteria, including N-fixing bacteria and mycorrhizal fungi, which further promote plant growth [63,67]. The organic carbon in vermicompost gradually releases nutrients into the root zone, allowing for steady nutrient absorption [65]. Moreover, vermicompost is crucial in producing plant growth regulators such as auxin and cytokinin by enhancing microbial communities and their activity in the root medium [68].

Our experiment demonstrated that both root and foliar applications of amino acids significantly promoted vegetative growth in tomato plants, leading to increased yield. Similar soil-grown studies with amino acids are consistent with the results obtained in our experiment [58,69,70]. Amino acids play essential roles in plants’ primary and secondary metabolism, participating in various enzymatic reactions, including those catalyzed by aminotransferases, dehydrogenases, lyases, and decarboxylases. As a result, they influence numerous phenological and physiological processes such as plant growth, seed germination, fruit ripening, stress response, water relations, photosynthesis, antioxidant capacity, nutrient absorption, and nitrogen storage [70,71].

Amino acid application has enhanced biochemical reactions in photosynthesis, increasing CO_2_ assimilation and promoting stomatal opening [58,72]. Specifically, applying aspartic and glutamic acids positively impacted the photosynthetic rate and stomatal conductance in tomato plants [58]. These amino acids also contributed to improvements in physiological and morphological parameters, partly through proline synthesis. Proline, a multifunctional amino acid, acts as an osmoprotectant, aids in osmotic adjustment, deactivates free oxygen radicals, regulates nutrient absorption, and enhances CO_2_ assimilation [73]. Furthermore, amino acid application has been reported to increase water use efficiency, chlorophyll content, and the gas exchange apparatus in tomato plants [70]. By promoting photosynthesis, amino acids likely enhance carbon production, boosting the plant’s redox potential and providing additional carbon and energy for growth [70].

PGPR inhabits the rhizosphere, thriving in, on, or around plant roots [74]. They contribute to improved plant performance by promoting growth, increasing yield, enhancing crop quality, and protecting against diseases and abiotic stress. The application of PGPR has been shown to improve growth parameters, photosynthetic efficiency, chlorophyll content, and yield in industrial tomatoes [75]. In our study, the application of PGPR in soilless tomato cultivation significantly enhanced plant growth and yield compared to control plants. The findings from our experiment align with similar soil-grown studies on the effects of amino acids in tomato plants [49,76,77,78]. PGPRs enhance plant nutrition through mechanisms such as biological nitrogen fixation, phosphorus solubilization, and the production of phytohormones such as auxins (IAA), gibberellins (GA), and salicylic acid (SA) [24,79]. They also facilitate nutrient uptake by producing ACC deaminase enzymes, synthesizing auxins, solubilizing nutrients via organic acids, and generating siderophores that chelate iron from the soil [77]. Moreover, the biochemical properties of PGPRs, including the production of amino acids, organic acids, and hormones, significantly boost nutrient absorption and overall plant growth.

Fulvic acids are soluble organic compounds widely present in nature and are essential components of organic matter, with the most negligible molecular weight among humic acids [80]. These compounds contain active functional groups capable of chelating and exchanging ions [33]. Fulvic acids support plant growth by enhancing membrane permeability, intracellular signaling, root development, chlorophyll levels, photosynthesis, and stimulating carbon and nitrogen metabolism [34]. They also provide essential amino acids, vitamins, trace elements, and hormones, promoting cell division, root growth, and nutrient uptake, thereby improving stress tolerance and crop yield [10]. In our study, the application of fulvic acid notably increased the growth and yield of soilless-grown tomato plants. Our experimental results agree with previous studies investigating the use of fulvic acids in tomato cultivation [34,81,82]. Studies have shown that spraying fulvic acids enhances crop growth [34,83] also reported that fulvic acids significantly improved seed germination, plant growth, and tomato yield in both soilless and soil-based systems, likely due to their promotion of root elongation and enhanced nutrient uptake, potentially linked to auxin-like substances in fulvic acids.

Chitosan has been found to promote growth across various plant species. In our study, chitosan enhanced tomato plants’ growth and yield to the control. The results of our experiment are supported by previous research on the application of chitosan in soil-grown tomatoes, as reported by Parvin et al. [84], Mondal et al. [85], and Reyes-Pérez Juan José et al. [86]. Chitosan was the least effective biostimulant in our experiment among those tested. El Amerany et al. [87] demonstrated that foliar application of chitosan improves tomato plant growth, leading to increases in leaf number, leaf area, and fresh and dry shoot weights compared to untreated plants. This growth enhancement is likely due to better chloroplast function, which boosts O_2_ production and CO_2_ fixation. Chlorophyll fluorescence (Fv/Fm) and stomatal conductance measurements indicated that chitosan facilitates stomatal opening, enhancing CO_2_ assimilation and photosynthetic activity.

### 4.2. Effects of Biostimulants on Fruit Quality Properties and Antioxidant Contents

In our study, PGPR emerged as the most effective treatment for enhancing the physical attributes of tomato fruits, making it the ideal choice for growers seeking larger and heavier produce [79,88]. The most significant improvements in PGPR-treated tomatoes were observed in fruit quality parameters: size, weight, diameter, titratable acidity, and total phenolic content. Additionally, these plants showed superior performance in other characteristics, including electrical conductivity (EC), vitamin C content, total soluble solids (TSS), and total flavonoids, compared to the control [74,77]. Katsenios et al. [75] reported that PGPR is associated with increased fruit weight, total carotenoids, phenolics, lycopene, antioxidant activity, and the activities of enzymes such as pectin methylesterase (PME) and polygalacturonase (PG) in industrial tomatoes. Other treatments related to fruit’s physical properties led to moderate improvements, but their effects were less pronounced than those of PGPR. Our findings also reveal a strong positive correlation between fruit weight, diameter, and volume. Treatments that increased fruit weight, such as PGPR, also resulted in larger fruit diameters and volumes, suggesting that these parameters are closely linked and likely influenced by the same growth factors [75].

The vermicompost enhanced the tomatoes’ red and yellow hues, creating a more vibrant and visually appealing fruit color. Other treatments, like amino acid and PGPR, also improved color parameters but not as significantly as vermicompost [89]. Chitosan-treated tomato fruits showed stronger pigmentation than those in the control [90].

All biostimulants increased skin elasticity compared to the control. The highest skin elasticity was observed in fruits treated with PGPR (8.70 kg cm^−2^). The increased elasticity suggests that the biostimulants may contribute to a more robust fruit skin, which could enhance resistance to mechanical damage during handling and storage.

Vermicompost resulted in the highest flesh firmness (3.67 kg cm^−2^), indicating firmer fruit flesh than other treatments. PGPR and amino acids also improved flesh firmness, showing a significant increase compared to the control and chitosan [89]. Firmer flesh may contribute to longer shelf life and better textural qualities in tomatoes, making the biostimulant’s treatments beneficial for post-harvest handling [88].

The vermicompost and PGPR treatments significantly enhanced the electrical conductivity of tomato fruits, indicating a potential increase in mineral nutrient accumulation, while the other treatments resulted in more moderate increases.

The PGPR and vermicompost treatments significantly enhanced the titratable acidity of tomato fruits, indicating a potential increase in flavor intensity. In contrast, fulvic acid and amino acid treatments increased moderately [91]. Chitosan, on the other hand, slightly decreased the titratable acidity.

All biostimulant treatments significantly increased TSS compared to the control, with the amino acid treatment being the most effective [89]. The treatments can be ranked from most to least effective in enhancing TSS as follows: amino acid > vermicompost > chitosan > fulvic acid > PGPR [90,91]. TSS primarily refers to the concentration of dissolved sugars, organic acids, and other soluble substances in tomato fruit [92].

The antioxidant results of our study indicate that all the treatments significantly enhanced the total phenolic content in tomato fruits, with PGPR treatment being the most effective [75]. Phenolic compounds are secondary metabolites widely recognized for their significant contribution to tomatoes’ nutritional and health-promoting properties [93]. These bioactive compounds are known for their potent antioxidant activity, which protects human cells from oxidative stress and related chronic diseases such as cardiovascular diseases, cancer, and neurodegenerative disorders [94]. The failure of high doses of synthetic antioxidants in pill form to prevent human diseases [95] underscores the importance of plant-derived antioxidants. This realization emphasizes the value of vegetables, highlighting their crucial role in our diet. Williams et al. [96] suggested that phenolic compounds could potentially impact cellular functions by targeting protein and lipid kinase signaling pathways. A thorough comprehension of how flavonoids function, either as antioxidants or as regulators of cell signaling, and the influence of their metabolism [97] on these functions is essential to assess their potential as potent biomolecules for preventing cancer, protecting the heart, and preventing neurodegenerative diseases [94,96].

All the treatments significantly boosted the total flavonoid content in tomato fruits compared to the control, with fulvic acid and amino acid treatments being particularly effective. These results highlight biostimulants’ potential to enhance tomatoes’ nutritional quality by increasing their flavonoid content [2,34,83].

To further emphasize the novelty of these findings, it is important to highlight the broader practical implications of the observed improvements in tomato fruit quality and nutritional properties. By specifically focusing on the nutritional content and antioxidant properties of soilless greenhouse tomatoes, this study not only contributes to understanding of how individual biostimulants impact these key parameters but also provides valuable insights into how growers can strategically utilize these substances to cultivate tomatoes with superior health-promoting characteristics.

The strong correlation between fruit size, weight, diameter, and volume in treatments such as PGPR suggests that these parameters are closely linked and influenced by the same growth factors. This highlights the potential for targeted use of biostimulants to improve marketable traits, which is significant for commercial growers aiming to enhance product quality. Furthermore, vermicompost’s ability to enhance color vibrancy, firmness, and elasticity indicates clear benefits in terms of both visual appeal and resistance to mechanical damage, which can improve shelf life during transport and storage.

The significant increases in total phenolics and flavonoid content across all treatments, particularly with fulvic acid and amino acids, reinforce the potential of biostimulants as natural enhancers of tomato antioxidant profiles. These bioactive compounds, known for their protective roles against oxidative stress and chronic diseases, further emphasize the health benefits of biostimulant-treated tomatoes, aligning with the rising demand for functional foods.

By improving both the nutritional and physical qualities of tomatoes, this study opens new opportunities for tailored agricultural practices aimed at improving both crop quality and consumer health benefits.

In this study, chitosan and vermicompost enhance the vitamin C content in tomatoes, highlighting their potential for improving the crop’s nutritional quality [84,86,90,98]. The increase in vitamin C due to specific biostimulants has significant implications, contributing to better antioxidant protection and immune function [2]. Vitamin C, including ascorbic acid and dehydroascorbic acid, is one of the primary nutritional traits in numerous horticultural crops. It possesses multiple biological functions in the human body and is commonly recognized as an antioxidant [99]. However, the physiological function of vitamin C is much broader. It facilitates iron absorption, produces hormones and carnitine, and plays crucial roles in epigenetic processes [99].

More than 90% of the vitamin C in human diets comes from fruits and vegetables, including potatoes [100]. According to Lee and Kader [100], various factors influence the vitamin C content of their products. The application of several biostimulants in our experiment showed that apart from amino acids, all other applications increased the content of vitamin C. Thus, biostimulants can act as a booster to mitigate pre-harvesting stress situations. A combination of nutrient uptake and stress tolerance enhancement leads to improved output quality and yield with economic benefits [101].

Chitosan treatment also significantly altered the fruit metabolome, increasing sucrose levels while decreasing organic acids, particularly citrate. It enhanced the concentrations of natural antioxidants, such as ascorbic acid, phytic acid, pantothenic acid, lycopene, and flavonoids. This increase in antioxidants is likely due to higher sugar production, which is converted into glucose-6-phosphate or other metabolites to alleviate oxidative stress induced by chitosan treatment. Thus, the elevated antioxidant levels in chitosan-treated fruits suggest that chitosan helps protect against cellular damage [87].

It is reported that foliar application of chitosan showed more significant benefits for fruit quality than leaves or roots. Metabolite analysis revealed that chitosan activated critical carbon and nitrogen metabolic pathways, improving CO_2_ fixation and increasing nitrogen and phosphorus levels, which led to higher sucrose production. This sucrose is a carbon source for synthesizing other metabolites, including phospholipids and antioxidants. In our study, chitosan-treated fruits exhibited superior physical and chemical properties and higher antioxidant content than control plants [87].

The findings of this study highlight the role of biostimulants in tomato nutritional quality. Vegetables maintain a balanced and healthy diet due to their high nutritional value, including secondary metabolites, vitamins, minerals, and dietary fiber. Additionally, they are low in calories and fat. Consuming vegetables has been linked to various health benefits, such as reducing the risk of chronic diseases and promoting overall well-being [102]. The positive impact of vegetables on health is thought to be due to the wide range of biological compounds rather than individual elements [102]. However, biostimulants represent only one of several key factors influencing the concentration of health-promoting compounds in vegetable produce. Other influential factors include genetic material, environmental and climatic conditions, agro-cultural practices, and harvesting methods. For example, high nitrogen fertilizer application generally reduces the vitamin C content in tomatoes, while reduced irrigation frequency can enhance it [102]. Additionally, practices such as bruising, mechanical damage, and excessive trimming negatively impact vitamin C retention [100]. Therefore, optimizing nutritional quality requires a coordinated approach integrating multiple plant growth aspects. Biostimulants can directly enhance nutritional quality, promote health compounds, and indirectly address critical challenges by reducing abiotic stress [19] without causing adverse environmental impacts.

## 5. Conclusions

This study provides valuable insights into the practical application of biostimulants in soilless greenhouse tomato cultivation. The novelty lies in evaluating their effects under controlled conditions, specifically targeting nutritional enhancement. These findings highlight the potential for practical applications in modern agricultural systems, offering growers innovative strategies to improve nutritional quality, apart from yield.

PGPR enhanced the physical and visual properties of soilless greenhouse tomatoes, particularly by increasing fruit size and weight. Additionally, PGPR significantly boosted total phenolic content, indicating potential antioxidant benefits. Vermicompost also contributed to higher levels of total phenolics, flavonoids, and vitamin C, thereby enriching the overall dietary profile of the tomatoes. It improved vital characteristics such as fruit color, firmness, and sweetness. Both PGPR and vermicompost significantly enhanced titratable acidity and electrical conductivity, further contributing to fruit quality. Meanwhile, amino acids and chitosan increased total soluble solids and vitamin C content, showcasing their role in improving tomatoes’ flavor and health-related properties.

Future research should focus on in-depth mechanistic studies of individual biostimulants to further elucidate their specific modes of action at the molecular and physiological levels. Additionally, investigating the synergistic effects of combining different biostimulants, such as PGPR and vermicompost, could reveal interactions that further optimize tomato quality. Examining how these biostimulants interact with genetic, environmental, and agronomic factors will be essential to fully leverage their potential in improving the nutritional value and overall productivity of soilless culture tomato systems.

## Figures and Tables

**Figure 1 plants-13-02587-f001:**
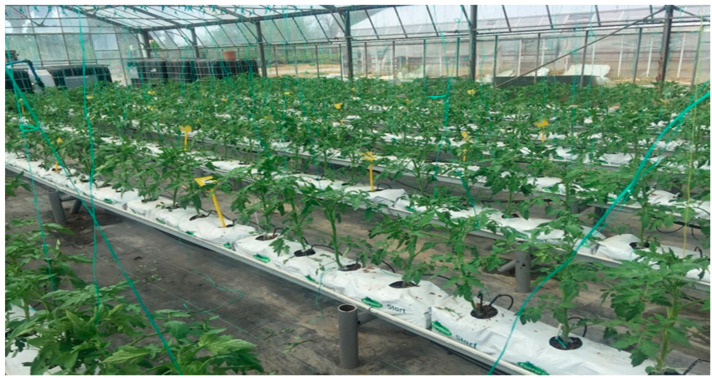
An image of tomato plants at the vegetative growth stage, 15 days after transplanting in coco coir slabs, shows biostimulants’ first application via foliar and root methods.

**Figure 2 plants-13-02587-f002:**
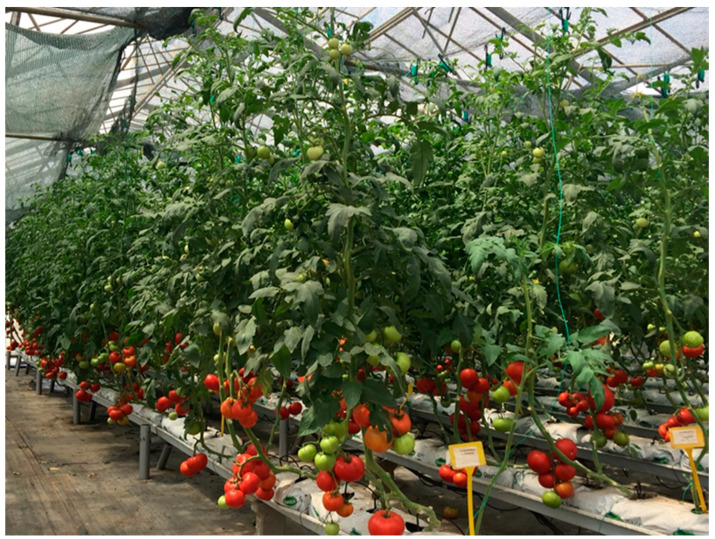
A view of tomato fruits that have reached the red ripening stage for harvest.

**Figure 3 plants-13-02587-f003:**
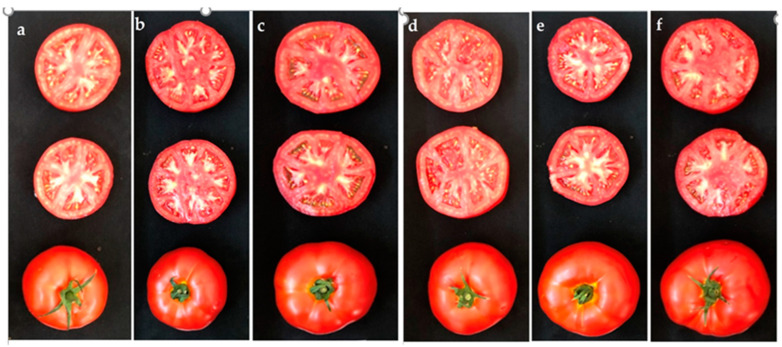
Images of soilless-grown tomato fruits with different biostimulants. (**a**): Control, (**b**): Amino acid, (**c**): PGPR, (**d**): Fulvic acid, (**e**): Vermicpompost, (**f**): Chitosan.

**Figure 4 plants-13-02587-f004:**
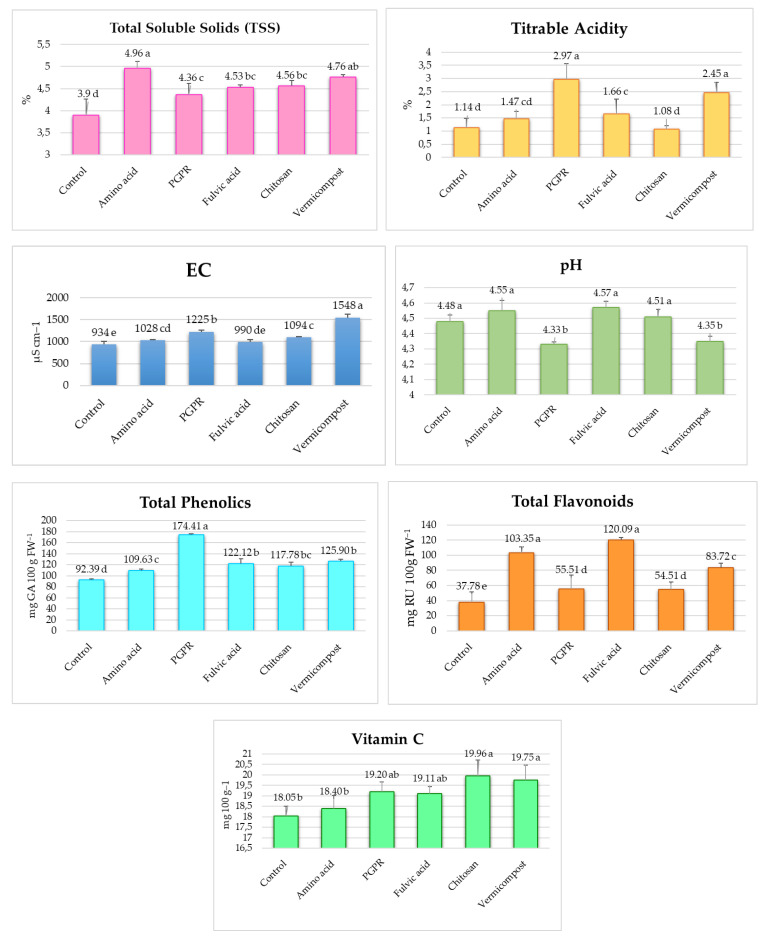
Effects of biostimulants on tomato fruit pH, EC (dSm^−1^), titratable acidity (mg 100 g FW^−1^), TSS (total soluble solids) (%), total phenolic compounds (mg GA 100 g FW^−1^), total flavonoid compounds (mg RU 100 g FW^−1^), and vitamin C (mg 100 g FW^−1^). There is no significant difference between means with the same letter in the same histogram; FW, fresh weight; GA, gallic acid; and RU, rutin.

**Figure 5 plants-13-02587-f005:**
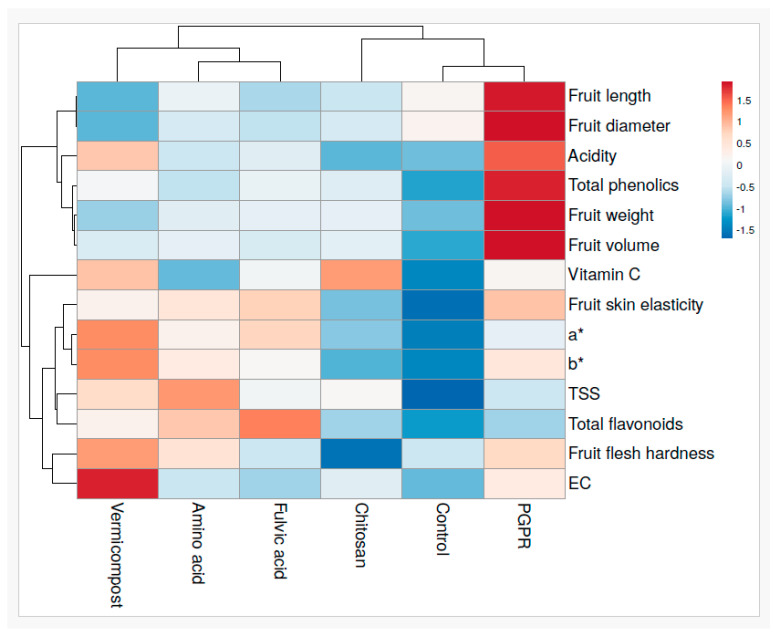
Heat map of tomato quality and nutritional attributes. Tomato fruit color parameters: a* (red green axis) measures the red green spectrum. b* (yellow blue axis) reflects the yellow blue spectrum.

**Figure 6 plants-13-02587-f006:**
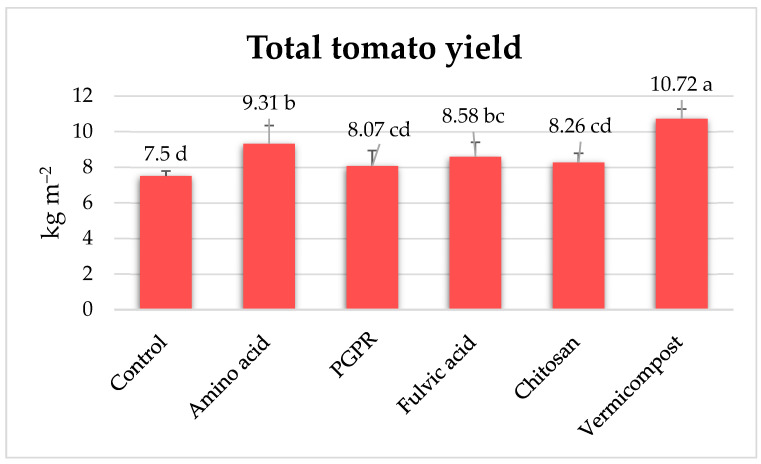
Effects of biostimulants on total tomato yield. In total, 7–8 clusters represented total yield and fruit number. There is no significant difference between means with the same letter in the same histogram.

**Figure 7 plants-13-02587-f007:**
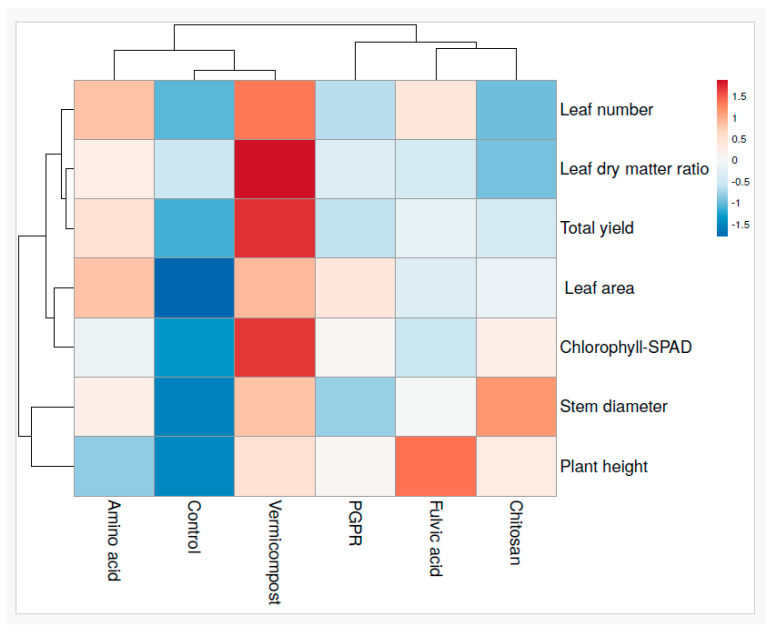
Heat map of tomato plant growth attributes and total fruit yield.

**Table 1 plants-13-02587-t001:** Doses of biostimulants used in the experiment applied via foliar and root treatments every 10 days.

Biostimulant	Root Application Dosage	Foliar Application Dosage
Amino acid	1.75 g L^−1^	0.6 g L^−1^
Benificial bacteria (PGPR)	1 mL L^−1^	3 mL L^−1^
Fulvic Acid	1.5 g L^−1^	1 g L^−1^
Chitosan	0.3 mL L^−1^	0.6 mL L^−1^
Vermicompost	2 mL L^−1^	3.5 mL L^−1^

**Table 2 plants-13-02587-t002:** Mineral fertilizers were utilized for the nutrient solution of soilless cultivated tomatoes.

Stock A	Stock B
Calcium nitrate	Potassium sulfate
Fe—EDDHA	Mono potassium phosphate
Potassium nitrate	Magnesium sulfate
	Microelements
	Zinc sulfate
	Boric acid
	Manganese sulfate
	Copper sulfate
	Ammonium molybdate

**Table 3 plants-13-02587-t003:** Impact of various biostimulant applications on growth parameters of soilless-grown tomato plants.

Treatments	Plant Height (cm)	Leaf Number per Plant	Leaf Area (cm^2^ Plant^−1^)	Stem Diameter (mm)	Leaf Dry Matter (%)	Leaf SPAD-Chlorophyll
Control	170 c	68.00 c	12,387 d	14.95 e	10.79 bc	37.13 d
Amino acid	173 bc	90.33 b	21,433 a	15.93 bc	11.55 b	44.00 bc
PGPR	177 b	73.00 c	19,916 ab	15.38 d	11.01 bc	44.90 bc
Fulvic acid	183 a	85.33 b	17,483 c	15.84 c	10.88 bc	41.42 c
Chitosan	178 ab	68.88 c	18,138 bc	16.48 a	10.39 c	45.70 b
Vermicompost	179 ab	96.66 a	21,600 a	16.29 ab	13.14 a	53.96 a
*p*	0.0011	0.0001	<0.0001	<0.0001	0.0023	<0.0001
LSD_0.05_	6.583	10.302	2157	0.419	1.062	4.07

LSD: the least significant difference between the means (*p* < 0.05). There is no significant difference between means with the same letter in the same column.

**Table 4 plants-13-02587-t004:** Impact of biostimulants on tomato fruit color characteristics.

Treatments	L	a	b
Control	40.94 a	26.70 c	32.45 c
Amino acid	37.40 cd	28.94 ab	37.38 b
PGPR	39.22 abc	28.46 abc	37.62 b
Fulvic acid	35.41 d	29.62 ab	36.69 b
Chitosan	39.62 ab	27.58 bc	33.63 c
Vermicompost	38.56 bc	30.32 a	39.92 a
*p*	0.0014	0.0191	<0.0001
LSD_0.05_	2.18	2.16	1.97

LSD: the least significant difference between the means (*p* < 0.05). There is no significant difference between means with the same letter in the same column.

**Table 5 plants-13-02587-t005:** Effects of biostimulants on the morphological characteristics of tomato fruits.

Treatments	Fruit Weight (g)	Fruit Length (mm)	Fruit Equatorial Diameter (mm)	Fruit Volume (cm^3^)	Fruit Skin Elasticity (kg cm^−2^)	Fruit Flesh Firmness (kg cm^−2^)
Control	164.74 c	36.91 b	57.81 b	146.62 c	6.31 b	2.91 bc
Amino acid	187.31 b	35.78 b	52.79 bc	173.38 b	8.31 a	3.37 ab
PGPR	257.49 a	47.07 a	73.23 a	228.45 a	8.70 a	3.46 ab
Fulvic acid	189.39 b	32.38 c	51.22 c	168.37 b	8.62 a	2.91 bc
Chitosan	189.74 b	33.07 c	52.94 bc	172.18 b	7.04 b	2.40 c
Vermicompost	169.79 c	30.36 d	47.45 c	169.05 b	8.07 a	3.67 a
*p*	<0.0001	<0.0001	<0.0001	<0.0001	0.0013	0.0188
LSD_0.05_	19.98	1.38	5.82	12.05	0.96	0.86

LSD: the least significant difference between the means (*p* < 0.05). There is no significant difference between means with the same letter in the same column.

## Data Availability

The data presented in this study are available in the article.

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
