# Peer review of "Biostimulants Enhance the Nutritional Quality of Soilless Greenhouse Tomatoes"

_plants, 2024, doi:10.3390/plants13182587_

Round 1
Reviewer 1 Report
Comments and Suggestions for Authors
The introduction is very extensive, I recommend reducing the information and placing what is most relevant and relevant to the work
line 158: in T3 you can handle only the abbreviation, and in the rest of the work, because there is no point in handling both
line 207: This is the first time I've read EC and it doesn't describe what it is.
line 267 have many brakets
line 312: use lightness from the first time it is mentioned and then only L*
line 339-354 nThey are very large paragraphs, they should be a maximum of 10 lines to focus the idea, organize the information or divide the paragraph
line 355-361 you do not need to repeat the results of the table
line 370-371, organize the parameters as mentioned in the methodology and thus mention the results
line 374 different size or font
line 397-410: gives results in graph and text. It is important not to give the same information. It is better to talk about increasing percentages and differences only, this for all these parameters. if anything the more and less
Lines 427 and 466 In the methodology I did not find any information about it (Heat map), how this analysis was carried out, or what program was used to carry it out. can go into statistical analysis
line 567 was talking about the application of chitosan not amino acids, then the bibliography is wrong because it does not refer to chitosan
The discussion is related to each of the treatments without relating them, however I do not read it as a conclusion as to why one of them was better compared to the other since in general it is mentioned how each one helped on the subject. It is important to highlight more what one can do and what the other does not do.
If the journal allows it, it would be better to put results and discussion together
If the magazine has no limit with the bibliography that is fine, but personally it seems excessive to me, since some are only mentioned once or in combination with others. Preferably, you should place those that contribute the most to the discussion of the work
Author Response
RESPONSES TO REVIEWER 1
Dear Editor and Reviewers,
Thank you very much for your valuable comments and suggestions. We accepted all your suggestions. The revised manuscript has been carefully updated, addressing the reviewers' comments point by point. The changes are highlighted in the manuscript in red colour. Please find our answer to your question below. At this stage, we would like to propose a change in the title of the manuscript. Instead of the current title, we would like to use the following new title "Biostimulants Enhance the Nutritional Quality of Soilless Greenhouse Tomatoes”. We believe that the new title will have a more significant impact and capture the attention of readers. We submit this matter to the discretion of the Editor and the reviewers.
Comment: The introduction is very extensive, I recommend reducing the information and placing what is most relevant and relevant to the work
Response: The introduction section has been shortened to the most relevant work, reduced from 105 to 95 lines, with a total reduction of 12.
Comment: line 158: in T3 you can handle only the abbreviation, and in the rest of the work, because there is no point in handling both
Response: Line 142: Only PGPR has been used in the revised manuscript.
Comment: line 207: This is the first time I've read EC and it doesn't describe what it is.
Response: EC refers to electrical conductivity, the solution’s ability to conduct electricity, which depends on the concentration of ions. Higher EC indicates more dissolved salts in nutrient solutions such as potassium, calcium, magnesium, etc., used for plant growth. For vegetable and fruit juices, EC varies with the type and concentration of acids and minerals, influencing the juice’s quality and preservation. EC is very commonly used in horticultural and plant sciences to assess nutrient levels. We wrote electrical conductivity (EC) when it was first mentioned.
Comment: line 267 have many brakets
Response: Line 251: Thank you very much for the revision. There was a typo, the brackets have been removed
Comment: line 312: use lightness from the first time it is mentioned and then only L*
Response Line 296: We have used "lightness" in the first mention and subsequently used "L*" throughout the manuscript.
Comments: line 339-354 They are very large paragraphs, they should be a maximum of 10 lines to focus the idea, organize the information or divide the paragraph
Response: Lines 317-326 and 327-331: Thank you for your suggestion. The manuscript has been revised to reduce the lines to 10 and 5, limited to the relevant issues.
Commnet: line 355-361 you do not need to repeat the results of the table
Response: Thank you. The lines 332-336 The numbers within the parentheses and those in the table have been removed.
Comment: line 370-371, organize the parameters as mentioned in the methodology and thus mention the results
Response: In the Methods section, lines 236-240, and the Results section, lines 343-381, as well as in the Figure, the tomato nutritional parameters have been organized according to the methodology specified, and the results have been reordered accordingly.
Comment: line 374 different sizes or font
Response: Lines 383-384 Different sizes or fonts have been adjusted.
Comment: line 397-410: gives results in graph and text. It is important not to give the same information. It is better to talk about increasing percentages and differences only, this for all these parameters. if anything the more and less
Response: Lines 367-375 have been revised as requested. The section has been rewritten with percentage increase values compared to the control.
Comment: Lines 427 and 466 In the methodology I did not find any information about it (Heat map), how this analysis was carried out, or what program was used to carry it out. can go into statistical analysis.
Response: Line 262-264 In the revised manuscript, the 'Statistical Analysis' section in the Methods part has been updated with the necessary information.
Comment: line 567 was talking about the application of chitosan not amino acids, then the bibliography is wrong because it does not refer to chitosan
Response: Thank you for the correction. There was a typographical error. The correction has been made by replacing “amino acid” with “chitosan” in line 528.
Comment: The discussion is related to each of the treatments without relating them, however I do not read it as a conclusion as to why one of them was better compared to the other since in general it is mentioned how each one helped on the subject. It is important to highlight more what one can do and what the other does not do.
Response: Thank you for your feedback. In this study, the discussion focuses on the effects of each biostimulant on the nutritional content of tomatoes individually. Each biostimulant has its own specific impact, and the effects of the biostimulants were evaluated independently.
Comment: If the journal allows it, it would be better to put results and discussion together
Response: Thank you for your suggestion. While we appreciate your input, we believe that keeping the Results and Discussion sections separate aligns with the journal's guidelines and is more appropriate for the clarity and structure of our manuscript. This separation allows for a more focused presentation of the data and a detailed interpretation of the results. We hope you understand our approach.
Comment: If the magazine has no limit with the bibliography that is fine, but personally it seems excessive to me, since some are only mentioned once or in combination with others. Preferably, you should place those that contribute the most to the discussion of the work
Response: Thank you for your valuable feedback regarding the reference list. We appreciate your concern about the number of references. However, each reference included in the manuscript serves a specific purpose, not only in the discussion but also in providing essential background information in the introduction and methodology sections. These references form the foundation of the study, supporting its depth and relevance. Reducing or altering the reference list could impact the overall context and comprehensiveness of the manuscript. That said, based on your suggestion, we have removed references 103, 104, and 105.

Reviewer 2 Report
Comments and Suggestions for Authors
The application of biostimulants in vegetable cultivation is becoming a promising way to promote crops productions in controlled environment agriculture and soilless culture systems. This study applied 5 kinds of biostimulants, such as amino acids, Plant Growth[1]Promoting Rhizobacteria (PGPR), fulvic acid, chitosan, and vermicompost along with mineral fertilizers to soilless greenhouse tomatoes in a coir pith medium. The experiments demonstrated that tomatos added with biostimulants performed better than control plants. PGPR enhanced the physical attributes of soilless greenhouse tomatoes, particularly in increasing fruit size and weight. Vermicompost raised the total phenolics, flavonoids, and vitamin C content, thereby enhancing the overall dietary profile of the tomatoes. These findings underline the specific nutritional benefits of different biostimulants, offering valuable insights for optimizing tomato cultivation practices to yield products with enhanced nutrition. The synergistic effects of combining different biostimulants is waiting for further study to optimize nutritional quality and yield in soilless tomato culture. The research results are useful for soilless greenhouse tomato production. The manuscript has done a comprehensive experiments and well organized and displayed for readers. It will be meaningful for scientific society.
However, I have a major concern for the research.
The biostimulants used in this study are commercial products, which indicates that their functions in agricultural crops have been fully studied and also is widely applied in production. Therefore, it is doubt that the innovative significance of researching on these biostimulants. By the way, this manuscript did not provide effective components and the composition of the biostimulants, making the results of this study dependent on specific commercial reagents. Therefore, scientific meaning is lack of broad value, and it is difficult to conduct in-depth mechanistic research.
Comments on the Quality of English LanguageIt is good for English writing.
Author Response
RESPONSES TO REVIWER 2
Dear Editor and Reviewers,
Thank you very much for your valuable comments and suggestions. Please find our answer to your question/comment below. At this stage, we would like to propose a change in the title of the manuscript. Instead of the current title, we would like to use the following new title "Biostimulants Enhance the Nutritional Quality of Soilless Greenhouse Tomatoes”. We believe that the new title will have a more significant impact and capture the attention of readers. We submit this matter to the discretion of the Editor and the reviewers.
1.Comments: The application of biostimulants in vegetable cultivation is becoming a promising way to promote crops productions in controlled environment agriculture and soilless culture systems. This study applied 5 kinds of biostimulants, such as amino acids, Plant Growth[1]Promoting Rhizobacteria (PGPR), fulvic acid, chitosan, and vermicompost along with mineral fertilizers to soilless greenhouse tomatoes in a coir pith medium. The experiments demonstrated that tomatos added with biostimulants performed better than control plants. PGPR enhanced the physical attributes of soilless greenhouse tomatoes, particularly in increasing fruit size and weight. Vermicompost raised the total phenolics, flavonoids, and vitamin C content, thereby enhancing the overall dietary profile of the tomatoes. These findings underline the specific nutritional benefits of different biostimulants, offering valuable insights for optimizing tomato cultivation practices to yield products with enhanced nutrition. The synergistic effects of combining different biostimulants is waiting for further study to optimize nutritional quality and yield in soilless tomato culture. The research results are useful for soilless greenhouse tomato production. The manuscript has done a comprehensive experiments and well organized and displayed for readers. It will be meaningful for scientific society.
1.Response: We appreciate your positive and encouraging comments. We also appreciate your recognition of our study's comprehensive nature and organization and its relevance to the scientific community. Your feedback is not only valuable but also motivates us to continue our research.
2.Comment: However, I have a major concern for the research. The biostimulants used in this study are commercial products, which indicates that their functions in agricultural crops have been fully studied and also is widely applied in production. Therefore, it is doubt that the innovative significance of researching on these biostimulants. By the way, this manuscript did not provide effective components and the composition of the biostimulants, making the results of this study dependent on specific commercial reagents. Therefore, scientific meaning is lack of broad value, and it is difficult to conduct in-depth mechanistic research.
2.Response: Thank you for your thoughtful feedback on our manuscript. While several biostimulants are commercially available and have been studied in various contexts, the novelty of our research lies in their specific application to soilless greenhouse tomato cultivation for increasing nutritional quality, which remains underexplored. We strongly believe our research contributes valuable information about their impacts on tomatoes' nutritional quality and physical attributes under controlled conditions.
Regarding the composition of the biostimulants, we would like to clarify that we have indeed provided detailed information about their active ingredients and functional components within the manuscript. This information is not proprietary or hidden but is included to ensure transparency and support our findings' validity. The first reviewer also highlighted this aspect and the paper's comprehensibility.
Furthermore, our study's approach to evaluating these well-established products under specific conditions contributes to the practical application of biostimulants in modern agricultural systems, which holds significant value for growers and researchers alike. We appreciate your suggestions and hope that our study’s contributions to both the practical implementation of biostimulants in soilless culture systems and the broader scientific understanding of their benefits are clear. Thank you for your consideration.
